# Toxic Encephalopathy and Methemoglobinemia after 5-Amino-2-(trifluoromethyl)pyridine Poisoning

**DOI:** 10.3390/ijerph192114031

**Published:** 2022-10-28

**Authors:** Yiming Tao, Longke Shi, Jie Han, Xiangdong Jian, Yongsheng Li

**Affiliations:** 1Department of Intensive Care Medicine, Tongji Hospital, Tongji Medical College, Huazhong University of Science and Technology, Hankou, Wuhan 430030, China; 2Department of Emergency, Tongji Hospital, Tongji Medical College, Huazhong University of Science and Technology, Hankou, Wuhan 430030, China; 3School of Public Health, Cheeloo College of Medicine, Shandong University, Jinan 250012, China; 4Department of Emergency, Qingdao Municipal Hospital, School of Medicine, Qingdao University, Qingdao 266071, China; 5Department of Poisoning and Occupational Diseases, Qilu Hospital of Shandong University, No. 107, Road Wenhuaxi, Jinan 250012, China

**Keywords:** 5-amino-2-(trifluoromethyl)pyridine, methemoglobinemia, acute renal failure, toxic encephalopathy, magnetic resonance imaging

## Abstract

The aromatic amino compound 5-amino-2-(trifluoromethyl)pyridine acts as an intermediate in the synthesis of pharmaceutical products. However, the toxicity profile of this compound is sparse and no related poisoning events have been reported. Here, we report the case of a 35-year-old man who inhaled 5-amino-2-(trifluoromethyl)pyridine at work. After inhalation, the patient rapidly developed symptoms such as dizziness, fatigue, nausea, vomiting, chest tightness, and loss of consciousness. After admission, methemoglobinemia, hemolytic anemia, acute renal failure, and toxic encephalopathy occurred. Symptoms improved significantly after intravenous treatment with a low dose of methylene blue. This revealed that 5-amino-2-(trifluoromethyl)pyridine is toxic to the human body and can be absorbed through the respiratory tract, resulting in methemoglobinemia and toxic encephalopathy; thus, caution should be taken in industrial production.

## 1. Introduction

The compound 5-amino-2-(trifluoromethyl)pyridine (C6H5F3N2; CAS Number: 106877-33-2) is a light yellow crystal. It causes severe eye, skin, and respiratory tract irritation as well as allergic skin reactions, or even death if ingested [1]. It acts as an intermediate in medicine and various chemical products [2,3]. However, to our knowledge, there are no systematic studies on the toxicology profile of 5-amino-2-(trifluoromethyl)pyridine, nor a relevant clinical profile. Here, we present a patient who was poisoned after inhaling 5-amino-2-(trifluoromethyl)pyridine and presented with methemoglobinemia, acute renal failure, and toxic encephalopathy.

## 2. Case Presentation

A 35-year-old male pharmaceutical laboratory technician with no relevant medical history had been working in a pharmaceutical plant for 10 years. The patient was assigned new tasks due to inadequate work force during the COVID-19 pandemic, thereby exposing him to 5-amino-2-(trifluoromethyl)pyridine for the first time. The patient did not often turn on the exhaust fan in the laboratory, did not wear respiratory protection equipment, and only wore rubber gloves. Moreover, the fume hood exhaust system ruptured, and vapor continued to leak into the laboratory during the extraction and recrystallization of 5-amino-2-(trifluoromethyl)pyridine. After three hours of continuous work, the patient developed symptoms such as dizziness, suffocation, nausea, vomiting, and tingling in the eyes. After two hours of rest, the patient resumed duty. After an hour of working again, the patient went into a coma and was found by a colleague in that state presumably after two hours. He was rushed to a local hospital for treatment. The patient briefly regained consciousness on the way to the hospital and was back in a coma about 20 min later. In the local hospital, endotracheal intubation with mechanical ventilation was performed, and the peripheral oxygen saturation dropped to 88%. He was then transferred to our hospital for treatment ten hours after the initial coma.

On admission, the patient’s lips and nail beds were dark brown, and his blood was chocolate-colored. The main laboratory test results were as follows: pH, 7.23; partial pressure of oxygen (PO_2_), 191 mmHg; oxygen saturation (SaO_2_), 81.70%; methemoglobin (MetHb), 39.40%; blood lactate, 5.6 mmol/L (Table 1). After injecting 100 mg methylene blue (MB) intravenously, within half an hour, the blood gas analysis results were as follows: pH, 7.33; SaO_2_, 93.30%; MetHb, 20.10%; blood lactate, 5.1 mmol/L. The nail bed and lips rapidly turned rosy. An hour later, the patient was given another intravenous bolus of 60 mg MB. Thereafter, 60 mg MB was administered intravenously daily for 4 days.

On the third day after onset, the patient recovered from coma. He had a mini-mental state examination (MMSE) score of 27, had no cognitive impairment, was off the ventilator, and was extubated. However, anuria, scleral jaundice, elevated indirect bilirubin, and a rapid decline in hemoglobin levels were observed on the same day. Based on this, the patient was judged to have hemolytic anemia. He was treated with dexamethasone 40 mg daily, continuous veno-venous hemofiltration (CVVH), and three plasmapheresis sessions (Figure 1). On the 10th day of onset, the urine output of the patient increased to 400 mL. Plasmapheresis and dexamethasone were discontinued, and CVVH therapy was continued.

During the course of treatment, the patient’s urine output increased. On the 19th day of onset, the patient’s urine output reached 2000 mL/day. However, at this time, his cognitive function declined, language comprehension and calculation ability declined, recent memory recall also decreased and he experienced mild disorientation, with a MMSE score of 19 and a Montreal cognitive assessment (MoCA) score of 15. The muscle strength of his appendages was graded at 5, with normal muscle tone and unremarkable tendon reflex and neck resistance, and a negative Babinski sign. Magnetic resonance imaging of the head revealed abnormal signals in the bilateral basal ganglia and corona radiata, and the patient was diagnosed with toxic encephalopathy. The lesion showed hypointense-to-isointense signal on T1-weighted images, and hyperintense signal on T2-weighted-images. Diffusion-weighted imaging (DWI), apparent diffusion coefficient (ADC) mapping and fluid-attenuated inversion-recovery (FLAIR) sequencing also demonstrated heterogeneous high signal (Figure 2).

The patient was given nerve growth factor to nourish the nerves, hyperbaric oxygen therapy, idebenone to improve cognitive function, and regular rehabilitation training. On the 25th day after onset, his laboratory test results were normal, prompting the patient’s discharge from the hospital and a continuation of citicoline sodium and idebenone at home. He continued rehabilitation training. Furthermore, 140 days after discharge from the hospital, the patient showed significant improvement in central nervous system symptoms and brain MRI results. However, he became more reticent than before the onset, and the others had recovered compared with those before the poisoning.

## 3. Discussing

To our knowledge, this is the first report of 5-amino-2-(trifluoromethyl)pyridine poisoning. Previously, few reports have described this compound. Its chemical structure is similar to that of aniline [2], which is oxidized into phenylhydroxylamine analogs after entering the human body, thereby causing methemoglobinemia, hemolytic anemia, abnormal liver and kidney function, and nervous system damage [4,5]. Our patient experienced similar symptoms, and laboratory findings confirmed the diagnosis of methemoglobinemia [6]. Therefore, based on its chemical structure and on the patient’s clinical manifestations, we hypothesized that 5-amino-2-(trifluoromethyl)pyridine oxidizes as hemoglobin to Met-Hb, as well as causing kidney and nervous system damage.

Methemoglobinemia occurs due to the oxidation of ferrous ions in erythrocytes to iron ions, resulting in excessive Met-Hb levels that cannot assure oxygen transport, thereby causing tissue hypoxia and damage [7,8]. Patients present with diffuse cyanosis and chocolate-colored arterial blood [9], and this hypoxia is difficult to ameliorate with oxygen. For these symptoms, MB is known to exert a curative effect. MB is an oxidant with two different effects on hemoglobin depending on its concentration in the body. At low concentrations, the hydrogen ions generated during glucose-6-phosphate dehydrogenation are transferred to MB through the reduction of pyridine nucleoside triphosphates, producing white methylene. Methylene white in turn reduces methemoglobin to normal hemoglobin [10]. High concentrations have the opposite effect, oxidizing normal hemoglobin to methemoglobin. Therefore, small doses (1–2 mg/kg) of MB should be given slowly intravenously during detoxification, as rapid and large doses will aggravate the symptoms of poisoning. Our patient quickly felt relieved of hypoxia following low-dose injection of methylene blue, showing that MB has a good effect on methemoglobinemia induced by 5-amino-2-(trifluoromethyl)pyridine. The patient developed hemolytic anemia and renal function damage due to tissue and cell hypoxia and damage caused by methemoglobinemia. Incidentally, 5-amino-2-(trifluoromethyl)pyridine is similar to aniline, which can reduce the production of reduced coenzyme II (NAD-PH) and inhibit the reduction of reduced glutathione (GSH), thereby rupturing red blood cells and causing hemolysis [11]. In such cases, high-dose steroids should be applied early on. However, if the condition cannot be effectively controlled even after high-dose steroid treatment, plasmapheresis can be performed to achieve good therapeutic effects [12,13].

On the 19th day of onset, the patient developed cognitive decline. MRI showed symmetrical hypointense signal on T1WI in the basal ganglia and corona radiata, and high signal on T2WI and DWI. The clinical and MRI findings were similar to those of delayed encephalopathy after acute carbon monoxide poisoning; however, the MRI findings are also similar to those of toxic encephalopathy caused by organic solvents including extensive white matter density reduction, symmetrical distribution, and no placeholder effect [14,15]. The chemical structure of 5-amino-2-(trifluoromethyl)pyridine is similar to aniline, which can cause neurological damage. However, the patient was comatose and experienced hypoxia for up to 10 h, and the possibility of delayed encephalopathy due to hypoxia could not be ruled out [16]. However, immediate removal, timely treatment of methemoglobinemia, and hypoxia correction are key to improve the overall patient prognosis, irrespective of the cause of toxic encephalopathy [6,17].

## 4. Conclusions

In conclusion, 5-amino-2-(trifluoromethyl)pyridine is toxic to the human body and can be absorbed through the respiratory tract; it can cause methemoglobinemia and hemolytic anemia. In our patient, both acute renal failure and delayed encephalopathy were observed. MB is effective in treating methemoglobinemia caused by 5-amino-2-(trifluoromethyl)pyridine poisoning. Early use of MB can treat methemoglobinemia and improve the patient prognosis.

## Figures and Tables

**Figure 1 ijerph-19-14031-f001:**
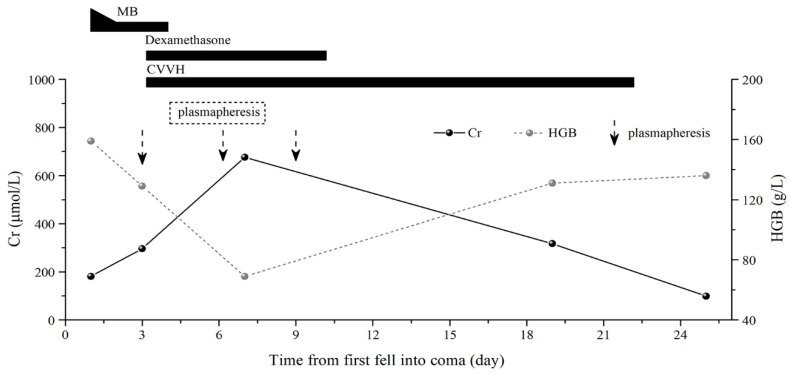
Clinical course of a patient with 5-amino-2-(trifluoromethyl)pyridine poisoning. MB, methylene blue; CVVH, continuous veno-venous hemofiltration.

**Figure 2 ijerph-19-14031-f002:**
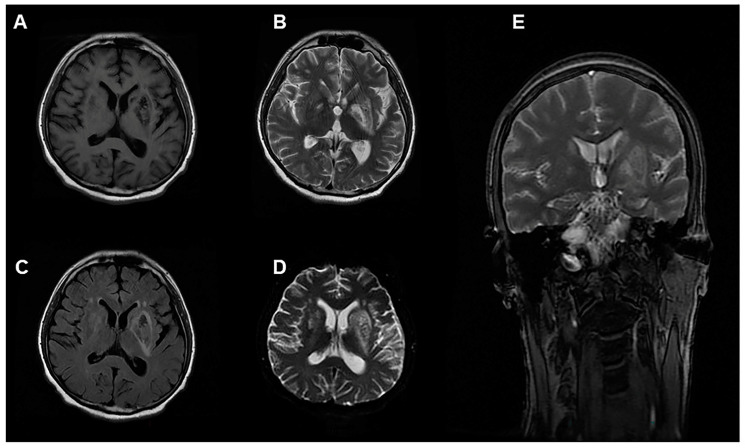
Images of a 35-year-old man exposed to 5-amino-2-(trifluoromethyl)pyridine. MRI 20 days after exposure showing abnormal signal intensity on (**A**) T1WI, (**B**) T2WI, (**C**) T2 FLAIR, (**D**) DWI and (**E**) T2WI of coronal in the bilateral cerebral hemispheres. T1WI, T1-weighted image; T2WI, T2-weighted image; DWI, diffusion-weighted imaging; FLAIR, fluid-attenuated inversion-recovery sequency.

**Table 1 ijerph-19-14031-t001:** Laboratory test results.

Parameters	10 h after Disease Onset	12 h after Disease Onset	3 Days after Disease Onset	7 Days after Disease Onset	19 Days after Disease Onset	25 Days after Disease Onset	Normal Values
WBC (×10^9^/L)	14.05	-	18.19	15.78	10.35	8.67	3.5–9.5
RBC (×10^12^/L)	5.26	-	4.54	2.78	4.19	4.22	4.3–5.8
Retic (×10^9^/L)				367.8	219.2	112.1	24–84
HGB (g/L)	159	-	129	69	131	136	130–175
PLT (×10^9^/L)	197	-	104	65	180	191	125–350
LDH (U/L)	3231		3973	2926	763	317	313–618
ALT (U/L)	96	-	142	127	23	25	9–50
AST (U/L)	101	-	181	136	19	17	15–40
DBIL (μmol/L)	7	-	9.1	13.1	2.7	2.1	0–5
IBIL (μmol/L)	18	-	27	35.3	11.8	9.7	0–19
BUN (mmol/L)	9.1	-	18.7	32.8	16.6	5.8	3.2–7.1
Cr (μmol/L)	181	-	296	676	317	99	57–97
PO_2_ (mmHg)	191	221	91	-	-	-	80–100
SaO_2_	81.70	93.30	96	-	-	-	>94%
MetHb	39.40	20.10	3.1	<1%	<1%	-	<1%

WBC, white blood cells; RBC, red blood cells; Retic, reticulocyte; HGB, hemoglobin; PLT, platelets; LDH, lactic dehydrogenase; ALT, alanine transaminase; AST, aspartate aminotransferase; BUN, blood urea nitrogen; Cr, creatinine; DBIL, direct bilirubin; IBIL, indirect bilirubin; PO_2_, partial pressure of oxygen; SaO_2_, percutaneous oxygen saturation; MetHb, methemoglobin.

## Data Availability

The original contributions presented in the study are included in the article, further inquiries can be directed to the corresponding author/s.

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
