# Peer review of "Toxic Encephalopathy and Methemoglobinemia after 5-Amino-2-(trifluoromethyl)pyridine Poisoning"

_ijerph, 2022, doi:10.3390/ijerph192114031_

Round 1
Author Response
Dear Reviewer 1:
Thank you very much for your comments. The article has benefited considerably from your review. We have carefully revised the manuscript according to your suggestions. We hope that the revised parts (highlighted in red) adequately address your concerns. Thank you very much for your insightful comments, which have significantly improved our manuscript.
Point 1:
Since the authors emphasized that the toxic effect was came after the skin exposure, the question I have here is what was the PH value of the chemical, and whether it was a liquid form or powder form, and how extensive of the exposure in term of Total Body Surface area?
Response 1:
We did not emphasize that the patient's intoxication was due to skin contact; in fact, the patient was wearing rubber gloves at all times (Page 2, lines 44), and had no skin exposure to 5-amino-2-(trifluoromethyl)pyridine. Following admission to the hospital, we carefully examined the skin of the patient’s entire body, and found no obvious corrosion-related injury. Perhaps, the word “exposure” in our abstract section was misleading (Page 1, lines19). Hence, we changed the word(Page 1, lines19). We offer our sincere apologies for not being clear enough in our description. Our original speculation was that the patient was exposed to an environment containing 5-amino-2-(trifluoromethyl)pyridine vapor, so the poisoning was caused by the inhalation of 5-amino-2-(trifluoromethyl)pyridine, not specifically skin exposure (Page 2, lines 35-36). We also contacted the patient to identify the pH of 5-amino-2-(trifluoromethyl)pyridine. The patient stated that the acidity coefficient (pKa) of 5-amino-2-(trifluoromethyl)pyridine was 1.49±0.22, but the specific pH value was related to the volume of the solution during recrystallization, and he had never examined it. Moreover, the patient also reconfirmed that the skin was not exposed to 5-amino-2-(trifluoromethyl)pyridine during the poisoning.
Point 2: As the authors described that obvious cyanosis of the earlobes and lips were noted in 30 minutes after the exposure, the quick development of the prominent symptoms and signs actually hinted that INHALATION may play more important role in causing such a significant clinical picture. Unless there was an extensive corrosive injury occurred on this case.
Response 2:
You are quite right. As you can judge, the patient was actually very likely to have inhaled 5-amino-2-(trifluoromethyl)pyridine. However, we do not know exactly when the cyanosis appeared in the patient. We have tried contacting the hospital where the patient was first admitted, and asked the patient to recall the time, however, we were unable to determine this information. What is certain is that upon entering our hospital, the patient’s lips and nail beds were dark brown, and his blood was chocolate-colored. At this point, the patient was in a coma for 10 hours (Page 2, lines 53-54), approximately 16 hours after the patient's first exposure to 5-amino-2-(trifluoromethyl)pyridine (Page 2, lines 44-47).
Point 3: If it was a significant corrosive injury, then the immediate first-aid of decontamination needs to be carried out on site without any delay (1-3 minutes) to prevent and/or to minimize the toxicities potentially associated.
Response 3:
We very much agree with you that early cleaning is very important for corrosion-based damage as well as certain pesticide poisonings. Once the patient had entered the hospital, we performed a careful examination of the patient's entire body skin, and found no trace of 5-amino-2-(trifluoromethyl)pyridine-related corrosion. Moreover, through inquiries with the patient's family members and co-workers, we speculated that the possibility of the patient's poisoning due to respiratory inhalation was very high. This was also confirmed by the patient himself once he was awake.
Point 4: Regarding the treatment, the authors mentioned that “Hemofiltration and intermittent hemoperfusion” was applied to treat the patient (on the 2nd day or so), I am curious about the rationale of these two procedures which had been exercised?
Response 4:
We employed both treatment modalities for 3 reasons:
- Blood perfusion has a better effect on removing macromolecular toxins from the blood(Vanholder Raymond,De Smet Rita,Glorieux Griet et al. Review on uremic toxins: classification, concentration, and interindividual variability.[J] .Kidney Int, 2003, 63: 1934-43); while hemofiltration has a better effect on removing medium and small molecular toxins(Friedrich Jan O,Wald Ron,Bagshaw Sean M et al. Hemofiltration compared to hemodialysis for acute kidney injury: systematic review and meta-analysis.[J] .Crit Care, 2012, 16: R146). Since this was the first case of poisoning by 5-amino-2-(trifluoromethyl)pyridine, there are multiple unanswered questions. For example, would this compound be transformed into new compounds within the human body? what would be the molecular weights of the new compounds? and what would be the protein binding rate? Regard of these unknowns, we expect a comprehensive and rapid removal of toxins from the body with hemoperfusion combined with hemofiltration.
- The patient developed severe hemolytic anemia and renal failure (Page 2, lines 70-72). Hemofiltration can quickly excrete certain toxic substances and metabolites from the body, and can improve the symptoms of hemolytic anemia(Andreoli Sharon Phillips,Acute renal failure.[J] .Curr Opin Pediatr, 2002, 14: 183-8.). At the same time, it can correct the water and electrolyte imbalance, and simultaneously reduce the serious adverse reaction of the patient. Given these efficacious properties, it is an effective treatment for acute renal failure(Ronco Claudio,Ricci Zaccaria,Renal replacement therapies: physiological review.[J] .Intensive Care Med, 2008, 34: 2139-46.).
- Following hemolysis, a large amount of bilirubin is generally produced due to hemoglobin decomposition. Elevated bilirubin not only causes jaundice in patients, but also is the main metabolite of iron porphyrin compounds. Bilirubin itself is toxic, and can cause irreversible damage to the brain and nervous system. Blood perfusion can effectively remove bilirubin, which is beneficial to the recovery of patients(Nocentini Alessio,Bonardi Alessandro,Pratesi Simone et al. Pharmaceutical strategies for preventing toxicity and promoting antioxidant and anti-inflammatory actions of bilirubin.[J] .J Enzyme Inhib Med Chem, 2022, 37: 487-501.).
Point 5: Clinically a delayed encephalopathy might develop secondary to any clinical situation of hypoxemia. I am wondering why if the authors choose to emphasize that the encephalopathy observed was more likely a feature of so-called toxic encephalopathy. Would the authors can specify?
Response 5:
You are correct that delayed encephalopathy may be secondary to clinical hypoxemia, which we also mentioned in the article (Page 5, lines 143-144). The patient's MRI findings were similar to those of hypoxic ischemic encephalopathy caused by carbon monoxide. However, the patient's MRI findings were also similar to brain damage caused by organic solvents (Page 5, lines 144-145). In fact, since this was the first report of 5-amino-2-(trifluoromethyl)pyridine poisoning in a patient who also experienced prolonged hypoxia, we were unable to determine which event (or both) played a major role in the patients' delayed encephalopathy. We also have no inclinations.
Definition of toxic encephalopathy: A disorder of the brain that alters the function or structure of the brain. It is typically caused by infectious agents (bacteria, viruses, or prions), metabolic or mitochondrial dysfunction, brain tumors or increased pressure on the skull, prolonged exposure to toxic elements (including, solvents, drugs, radiation, paints, industrial chemicals, and certain metals), chronic progressive trauma, malnutrition, hypoxia, or insufficient blood supply to the brain(https://www.uptodate.com/contents/acute-toxic-metabolic-encephalopathy-in-adults#H11068804). Previous studies attributed hypoxic encephalopathy caused by substances, such as, carbon monoxide to class for toxic encephalopathy(Dietemann J-L,Botelho C,Nogueira T et al. [Imaging in acute toxic encephalopathy].[J] .J Neuroradiol, 2004, 31: 313-26). Therefore, we hold the opinion that either delayed encephalopathy caused by hypoxia induced by 5-amino-2-(trifluoromethyl)pyridine, or brain damage caused by 5-amino-2-(trifluoromethyl)pyridine and its metabolites, both can be classified as toxic encephalopathy. Therefore, we used the term toxic encephalopathy on (Page 5, lines 150-152). However, that does not mean that we are more inclined to a certain etiology. In fact, your question is of great significance, and more clinical cases and basic research are needed to answer this question, which is also the focus of our next work.
Once again, I would like to express my sincere thanks to you. Your suggestions have been of great help and have made us sort out the structure of our article. We look forward to hearing from you regarding our submission. We would be glad to respond to any further questions and comments that you may have.

Reviewer 2 Report
This article describes a case with toxic encephalopathy and methemoglobinemia after 5-amino-2-(trifluoromethyl)pyridine poisoning. Although this the first report of 5-amino-2-(trifluoromethyl)pyridine poisoning, because this substance acts as an intermediate in medicine and various chemical products, it is very important to address this case report. This article is written very well and the presentation flow is smooth.
The author presents a diffusion-weighted image on brain MRI. To better define the restricted diffusion pattern of the basal ganglia lesion, please show the aparent diffusion coefficient (ADC) mapping image in figure or mention about the ADC findings in maintext.
Author Response
Dear Reviewer 2:
It is our great honor to have your affirmation. We appreciate and have incorporated your suggestions, which have certainly strengthened the manuscript.
Point 1:
The author presents a diffusion-weighted image on brain MRI. To better define the restricted diffusion pattern of the basal ganglia lesion, please show the aparent diffusion coefficient (ADC) mapping image in figure or mention about the ADC findings in maintext.
Response 1:
Your suggestion is very helpful. With the ADC image, we can tell if the water molecules are truly diffusion limited. However, owing to the version problem of the MRI system, we were unable to export high-definition ADC images, and therefore, we only provided rephotograph images via mobile phones. Therefore, the image resolution was low, and failed to meet the requirements for publication. We are very sorry about this. However, it can still be seen that the lesions exhibited high signal on the ADC image, which was similar to the results of the toxic encephalopathy caused by organic solvents(Zheng Q N,Sheng W S,Pan A S,[Analysis of 15 cases of toxic encephalopathy caused by acute benzene poisoning].[J] .Zhonghua Lao Dong Wei Sheng Zhi Ye Bing Za Zhi, 2022, 40: 694-697). This result has also been added to the text(Page 3 lines93).
Once again we would like to express our heartfelt thanks to you. We look forward to hearing from you regarding our submission. We would be glad to respond to any further questions and comments that you may have.
(For pictures, please check the word file I sent.)

Reviewer 3 Report
The authors clearly and shortly reported the case, including not only sufficient details regarding clinical presentation but also explaining the mechanisms of agent and antidote actions. If the present study was accepted, it would be significant contribution in the field of clinical toxicology.
The acute poisoning described in the paper is caused by the aromatic agent 5-amino-2-(trifluoromethyl)pyridine. This substance has been blamed for acute poisoning but just as an irritative agent, able to provoke irritative signs and symptoms on the skin and mucouse membranes of the exposed patients. The authors for the first time reported systemic effects of the mentioned poison. The first sign of poisoning is methemoglobinemia, which was successfully treated/reversed with the specific antidote methylene-blue, while subsequent renal failure and encephalopathy could be provoked by direct toxic effect of the agent but also could be a consequence of tissue and cell hypoxia, as the authors mentioned as well. As it has been well established that anilin can be responsible for development of methemoglobinemia, haemolytic anaemia and kidney and brain toxicity/damage, the authors underlined/revealed that structural similarity between 5-amino-2-(trifluoromethyl)pyridine and aniline is a key of 5-amino-2-(trifluoromethyl)pyridine mechanism of toxicity.
Very interesting report, which will be of a great interest for the clinical toxicologist worldwide to successfully treat acutely poisoned patients by 5-amino-2-(trifluoromethyl)pyridine, as the authors in great details, step by step, explained treatment options. Also, a table, a figure and a picture in the text support all was written.
Author Response
Dear Reviewer 3:
It is our great honor to have your affirmation. We appreciate your review, which have certainly strengthened the manuscript.
Point 1:
Very interesting report, which will be of a great interest for the clinical toxicologist worldwide to successfully treat acutely poisoned patients by 5-amino-2-(trifluoromethyl)pyridine, as the authors in great details, step by step, explained treatment options. Also, a table, a figure and a picture in the text support all was written.
Response 1:
Once again we would like to express our heartfelt thanks to you. We would be glad to respond to any further questions and comments that you may have.
